# A Comparative Study of Lightweight, Sparse Autoencoder-Based Classifiers for Edge Network Devices: An Efficiency Analysis of Feed-Forward and Deep Neural Networks

**DOI:** 10.3390/s25206439

**Published:** 2025-10-17

**Authors:** Mi Young Jo, Hyun Jung Kim

**Affiliations:** 1Engineering Education Research Center, Konkuk University, Seoul 05029, Republic of Korea; miyoung2@konkuk.ac.kr; 2Sang-Huh College and the Graduate School of Information & Communication, Konkuk University, Seoul 05029, Republic of Korea

**Keywords:** edge computing, sparse autoencoder, lightweight classifier, network anomaly detection, feed-forward neural network, deep neural network, CIC-IDS2017, Edge Performance Efficiency Score (EPES)

## Abstract

This study proposes a lightweight classification framework for anomaly traffic detection in edge computing environments. Thirteen packet- and flow-level features extracted from the CIC-IDS2017 dataset were compressed into 4-dimensional latent vectors using a Sparse Autoencoder (SAE). Two classifiers were compared under the same pipeline: a Feed-Forward network (SAE-FF) and a Deep Neural Network (SAE-DNN). To ensure generalization, all experiments were conducted with 5-fold cross-validation. Performance evaluation revealed that SAE-DNN achieved superior classification performance, with an average accuracy of 99.33% and an AUC of 0.9993. The SAE-FF model, although exhibiting lower performance (average accuracy of 93.66% and AUC of 0.9758), maintained stable outcomes and offered significantly lower computational complexity (~40 FLOPs) compared with SAE-DNN (~8960 FLOPs). Device-level analysis confirmed that SAE-FF was the most efficient option for resource-constrained platforms such as Raspberry Pi 4, whereas SAE-DNN achieved real-time inference capability on the Coral Dev Board by leveraging Edge TPU acceleration. To quantify this trade-off between accuracy and efficiency, we introduce the Edge Performance Efficiency Score (EPES), a composite metric that integrates accuracy, latency, memory usage, FLOPs, and CPU performance into a single score. The proposed EPES provides a practical and comprehensive benchmark for balancing accuracy and efficiency and supporting device-specific model selection in practical edge deployments. These findings highlight the importance of system-aware evaluation and demonstrate that EPES can serve as a valuable guideline for efficient anomaly traffic classification in resource-limited environments.

## 1. Introduction

### 1.1. Background and Motivation

With the explosive growth of data generated by the proliferation of sensors and IoT devices, transforming raw data into actionable information has become a key research challenge. Recent studies have demonstrated the feasibility of on-device intelligence by embedding AI modules directly into wireless smart sensors to classify collision events locally, proving the practical potential of edge computing. These developments underline the growing convergence among IoT, cloud, edge computing, and AI technologies [1,2,3,4].

Edge computing reduces latency and enables real-time services by processing data and performing computations at the network edge where data is generated [5]. However, edge devices such as the Raspberry Pi and Jetson Nano are subject to physical constraints, including limited memory capacity, computational performance, and power [6]. These inherent characteristics of edge computing have driven research into real-time applications and resource management strategies suitable for such environments [7,8,9].

In these resource-constrained settings, tasks like network traffic anomaly detection must be performed in real-time, even on low-power and low-specification hardware. Various deep learning models, including Deep Neural Networks (DNNs), Recurrent Neural Networks (RNNs), Convolutional Neural Networks (CNNs), Autoencoders, and Graph Neural Networks (GNNs), have been utilized to classify real-time traffic. Nevertheless, there is a persistent need for lightweight models that can maintain high accuracy while minimizing latency [10,11,12,13]. To overcome these limitations, researchers have explored model optimization techniques, Autoencoder-based feature extraction, and specialized lightweight architectures like MobileNet [14], EfficientNet [15], and TinyML [16].

Recent work has further advanced hybrid AI frameworks for mobile and edge networks, where computation is split between the edge and the cloud [17,18,19]. For example, integrating K-Nearest Neighbors (KNN) on the edge with Long Short-Term Memory (LSTM) on the cloud has been shown to reduce delay, energy, and bandwidth consumption by 35%, 28%, and 60%, respectively. These results emphasize the importance of evaluating trade-offs between accuracy and efficiency—the core objective of this study.

In addition, recent advances in edge-optimized neural architectures and hardware-aware model design—including model sparsification, quantization, knowledge distillation, and neural architecture search—have contributed to improving efficiency under device constraints. More specifically, recent works such as [20] highlight that hardware-aware approaches, which consider the specifics of the hardware architecture during the model design phase, are emerging as a key strategy for maximizing performance in edge AI. Yet, many of these studies focus on optimizing a single metric (e.g., accuracy or latency), leaving a gap in unified evaluation frameworks that capture the overall trade-off. Indeed, recent surveys [21] have also identified the lack of a unified metric for fairly evaluating model deployment suitability in heterogeneous edge environments as a key future challenge. To fill this gap, we introduce the Edge Performance Efficiency Score (EPES), which provides an integrated metric for assessing accuracy, latency, FLOPs, and power in practical edge-AI deployments.

In this paper, we utilize a Sparse Autoencoder (SAE) [22], an unsupervised deep learning model, to compress and extract the core features from network traffic and flow data. The SAE model selectively activates input data to learn a sparse representation, thereby extracting the most significant feature vectors. We then conduct a comparative analysis of two lightweight classifier architectures—a FF model and a DNN model—which take the SAE-compressed vectors as input. These classifiers, despite their simple network structures, demonstrate high classification performance, making them well-suited for constrained environments like edge computing. Notably, this study moves beyond a simple accuracy comparison to provide a comprehensive evaluation of the models’ feasibility on edge devices by considering inference time, FLOPs, memory usage, and power consumption.

### 1.2. Related Work

#### 1.2.1. Feature Extraction Based on Sparse Autoencoders

Various deep learning-based feature extraction methods have been applied to detect abnormal behavior in network traffic. Among these, the SAE is an unsupervised learning model that performs efficient compression while preserving the main features of the input data. SAE encourages sparsity by keeping most hidden neurons inactive while selectively activating only a few, thereby learning a sparse representation.

As shown in Equation (1), the SAE encodes an input vector (*x*) into a compressed hidden representation and reconstructs it ($\hat{x}$). The loss function consists of the Mean Squared Error (MSE) with an additional Kullback–Leibler (KL) divergence term to enforce sparsity (ρ). In this formulation, β is the weight coefficient of the sparsity term, and ρ denotes the average activation of hidden units, which is constrained to reach the target sparsity level.(1)$$L(x,\hat{x})=MSE(x,\hat{x})+\beta \;kL(\rho ∥\hat{\rho })$$

A number of studies have applied SAE to network intrusion detection and traffic classification. For example, Ref. [23] proposed a semi-supervised stacked autoencoder that mitigates overfitting and improves robustness by using denoising and dropout. The stacked Denoising Autoencoder (sDAE) [24] extends this idea by injecting noise into the input and training the model to reconstruct it, thereby learning deeper representations. Similarly, Ref. [25] proposed the stacked Pruning Sparse Denoising Autoencoder (sPSDAE), which prunes less important units and combines denoising with sparsity to improve training efficiency.

Traditional dimensionality-reduction techniques such as Principal Component Analysis (PCA) and Linear Discriminant Analysis (LDA) are effective for simplifying data but often struggle to capture the non-linear relationships inherent in complex traffic patterns. Methods like t-distributed Stochastic Neighbor Embedding (t-SNE) and Uniform Manifold Approximation and Projection (UMAP) provide strong capabilities for visualizing hidden data structures; however, their non-parametric and stochastic nature makes them less suitable for real-time edge inference, as they cannot consistently map newly arriving data [26,27,28].

This study utilizes an SAE to capture the non-linear characteristics of network traffic while maintaining a sufficiently low computational burden even in lightweight environments. SAE effectively transforms raw packet and flow data into compressed latent features through a learnable encoder, which are then used as input for two classifiers: an FF and a DNN, which are evaluated in the following sections.

While many existing studies [29,30,31] have focused on improving the accuracy of autoencoder-based feature extraction, a systematic comparison of lightweight classifiers using SAE-derived features or a comprehensive review of their suitability for edge deployments remains lacking. To address this gap, this study utilizes SAE as a common feature extractor and directly compares FF and DNN within the same pipeline. Furthermore, we introduce the EPES, a composite metric designed to evaluate model suitability for edge devices by considering not only accuracy but also latency, FLOPs, memory usage, and power consumption.

#### 1.2.2. Lightweight Deep Learning in Edge Computing Environments

In network traffic analysis, one important factor is that anomaly detection must be performed in environments with limited resources. Therefore, the choice of deep learning models should be appropriate for the computing environment. Research on lightweight deep learning models has been conducted, including MobileNets, EfficientNet, and TinyML.

MobileNets, developed by Google, were designed for mobile and embedded vision applications. To optimize computational efficiency, only the first layer uses a standard convolution, while the remaining layers adopt depthwise separable convolutions. This design reduces computation by approximately 8–9 times compared with standard convolutions. In addition, two hyperparameters—the width multiplier (which uniformly reduces the number of channels in each layer) and the resolution multiplier (which adjusts the input image resolution)—make it possible to operate under resource-constrained environments.

EfficientNet is a study on scaling convolutional neural networks, proposing a compound scaling method that balances depth, width, and resolution simultaneously. Compared with models such as MobileNets, EfficientNet has demonstrated significant performance improvements even when the compound scaling approach is applied under the same conditions.

TinyML focuses on implementing machine learning techniques in extremely small devices such as microcontrollers, where memory, computation, and power are highly limited. To address these constraints, model compression techniques such as pruning, quantization, and Huffman coding are used, along with architectural optimization methods including depthwise separable convolution and knowledge distillation. These techniques aim to minimize hardware requirements while maintaining model effectiveness.

These approaches show that in edge computing environments, lightweight deep learning models are essential to handle traffic analysis and anomaly detection under resource limitations.

#### 1.2.3. Research in Anomaly Detection

Deep learning–based approaches to network anomaly detection have been developed in various forms, including prediction-based, reconstruction-based, representation learning–based, and hybrid methods [32]. In edge environments, several studies have focused on reducing data dimensions and selectively processing only part of the data (horizontal reduction), combined with the use of RNNs for training [33]. Other research has compared different autoencoders, such as AE, SAE, VAE, and adversarial AE, for unsupervised anomaly detection [31]. The ECADA framework [8] proposed an edge–cloud collaborative method that updates and distributes rule-based anomaly detection (ARB). In addition, surveys on deep anomaly detection in multivariate time series [34] have summarized recent developments.

Many of these studies, however, tend to report only accuracy or a specific efficiency indicator in a partial manner. In summary, the majority of prior work can be categorized into three tendencies:focusing on improving representation learning accuracy with SAE,reporting only a single efficiency aspect, such as latency or power consumption, andevaluating models under fixed datasets or hardware conditions with a limited scope [35].

As a result, a systematic framework for evaluating edge deployment suitability by integrating accuracy and efficiency into a single indicator has not yet been fully established.

#### 1.2.4. Distinctiveness and Contributions

In this study, we use the CIC-IDS2017 [36] dataset to construct a 13-dimensional feature vector (8 from packet headers and 5 from flow statistics) and train an SAE to learn a low-dimensional latent vector. This vector is then fed into the FF and DNN classifiers. The models are evaluated using 5-fold cross-validation, and the inference performance was simulated based on the official specifications of three representative edge devise: Raspberry Pi 4, Jetson Nano, and Coral Dev Board. Performance is measured using standard metrics like Accuracy, F1-score, and AUC. To quantitatively assess their suitability in edge environments, we introduce a novel, comprehensive metric called the EPES, which incorporates inference time, memory usage, FLOPs, and device-specific performance characteristics. Our analysis reveals that while the DNN model achieves superior accuracy, the much simpler FF model provides a more practical balance of performance and efficiency, highlighting a critical trade-off for real-world edge deployment.

This study addresses the limitations mentioned above and summarizes its explicit contributions as follows:Systematic comparison of SAE-FF and SAE-DNN within the same pipeline:

Using the CIC-IDS2017 dataset, both classifiers are compared under identical preprocessing, feature extraction (SAE), training, and evaluation protocols. Their performance is analyzed in terms of accuracy (Accuracy/AUC) and efficiency (latency, FLOPs, memory, and power) across different devices.

Proposal of a unified metric for edge deployment suitability (EPES):

We introduce the EPES, which combines normalized accuracy and efficiency indicators into a single measure. This allows quantitative support for deployment decisions in edge environments simulated by representative devices such as Raspberry Pi 4, Jetson Nano, and Coral Dev Board.

Enhanced reproducibility through combined measurement and modeling:

For inference latency and power consumption on each device, both direct measurements (when feasible) and modeling are reported. In addition, sensitivity analysis of weighting factors is conducted to verify the practical applicability of EPES.

In designing the experiment, the preprocessing steps, evaluation metrics, and device configurations were selected to reflect realistic constraints often faced in real-world edge environments—particularly latency, power, and memory limitations. These choices were intended to make the evaluation more representative of actual deployment conditions for lightweight network intelligence.

Overall, the proposed framework assesses model suitability for edge environments from a holistic perspective through the EPES metric. Table 1 summarizes the scope of the evaluation and its comparison with related studies (see Section 3.2 for the detailed definition and calculation of EPES).

## 2. Methods

### 2.1. Overall System Architecture

This study compares lightweight classifiers that use features extracted by an SAE to classify network packets from the CIC-IDS2017 dataset. The system operates as a multi-step pipeline.

(Step 0. Preprocessing): In this stage, PCAP files are sampled and CSV files are cleaned.(Step 1. Feature Abstraction): The SAE extracts latent features from the preprocessed data.(Step 2. Classification): A lightweight classifier—either an FF network or a DNN—uses the latent features to label traffic as normal (BENIGN) or abnormal (ATTACK).

The overall architecture of the SAE-FF/DNN models, shown in Figure 1, has four stages: data preprocessing, feature extraction with SAE encoding, classification by FF or DNN, and inference on edge devices.

The process starts by creating a 13-dimensional feature vector. Eight features come from packet headers in the PCAP files, and five are flow statistics taken from the CSV files. This vector is passed to the SAE encoder. A sparsity constraint of 0.05 and a KL divergence penalty (β = 1 × 10^−3^) are applied to produce a compact latent representation with minimal information loss. These latent features then serve as input to the FF and DNN classifiers. Both models are designed with lightweight structures by minimizing the number of hidden-layer neurons. They perform binary classification, identifying each traffic flow as either normal or abnormal. To test their real-world applicability, experiments and performance simulations were carried out on resource-limited edge devices: Raspberry Pi 4, Jetson Nano, and Coral Dev Board.

### 2.2. Edge Performance Efficiency Score

This study introduces the EPES to provide a unified metric for evaluating the deployability of models on edge devices. EPES consolidates both classification performance and computational efficiency into a single value, enabling fair comparison between models such as FF and DNN when applied in constrained environments. The score comprises five components—accuracy, inference speed, FLOPs, memory efficiency, and CPU performance factor—each normalized and weighted according to their relative importance (see Table 2 for details).

*Accuracy Score*: Accuracy reflects the model’s fundamental ability to correctly classify benign and malicious traffic. The average accuracy from 5-fold cross-validation is directly applied, and because accurate detection is the most critical objective, this factor is weighted at 25%.

*Speed Score*: Inference latency is a key determinant of real-time applicability. If the inference time is faster than 10 ms, the score is set to 1.0. Otherwise, it is normalized using the function Min (1.0, 10.0/inference time (ms)). Given the criticality of low latency in edge computing, this score is also weighted at 25%.

*FLOPs Score*: FLOPs measure the floating-point operations required for a single inference and represent computational complexity. A lower FLOPs count reduces latency and energy consumption, justifying a 20% weight.

*Memory Efficiency*: Memory usage accounts for both model size and runtime data requirements. Since edge devices typically operate under strict memory constraints, memory efficiency is weighted at 15%.

*CPU Performance Factor*: To ensure fair comparison across devices with varying architectures and clock frequencies, the relative performance of each CPU is normalized against a baseline (Intel i7-7700K, 4.2 GHz). This factor, weighted at 15%, supplements the other metrics by adjusting for architectural differences.

The overall EPES is computed as:$$EPES=\left(0.25\;\times Accuracy\right)+\left(0.25\;\times Speed\right)+\left(0.20\times \;FLOPs\right)\;\;+\left(\;0.15\;\times Memory\right)+(0.15\times \;CPU\;Performance)$$

Energy consumption was additionally estimated as the product of device power (W) and inference time (s), though this was not directly integrated into the EPES formula. Model size was excluded, as both FF and DNN had sizes in the kilobyte range, providing limited discriminative value.

### 2.3. SAE-Based Feature Compression

In this study, we apply a SAE, a type of unsupervised learning model, to compress a 13-dimensional input feature vector into a 4-dimensional latent space. The goal is to capture the essential characteristics of network traffic while discarding redundant information. By reconstructing the input from these latent variables, the model achieves dimensionality reduction with minimal information loss. The resulting compressed features are then used as inputs to the classifiers, which improves both training speed and inference efficiency [22].

The input vector consists of 13 features: 8 extracted from packet headers in the PCAP files and 5 derived from flow-level statistics in the CSV files. We tested hidden layers with 3, 4, 6, and 8 dimensions. Among these, the 4-dimensional configuration was selected [31], as it maintained a classification accuracy of about 93.66% while providing the most compact representation.

To further refine the SAE, we controlled the sparsity of the hidden layer so that the average activation remained below 5%. This level falls within the commonly recommended range of 1–10% [37]. We evaluated several sparsity values (0.01, 0.03, 0.05, and 0.07) and found that setting it to 0.03 offered the best trade-off between classification accuracy and reconstruction loss. In addition, a KL divergence penalty term (β = 1 × 10^−3^) was applied to reinforce the sparsity constraint. For activation, we used LeakyReLU (α = 0.1) [38] instead of the standard ReLU to achieve more stable training. Finally, to prevent overfitting, a Dropout [39] of 20% (*p* = 0.2) was applied to the encoder’s output, a rate consistent with the 10–30% range generally recommended for SAE architectures.

### 2.4. Classifier Design

#### 2.4.1. Feed-Forward Classifier

The FF classifier is designed as a lightweight model that operates on the latent features derived from the SAE, as illustrated in Figure 2. By using these pre-compressed features, the model is designed to be a simple yet computationally efficient binary classifier.

In this study, the SAE-FF architecture was chosen as a lightweight baseline within the proposed framework. This configuration allows us to examine how well the SAE’s latent features can be classified using a minimal network structure. Because the FF classifier contains very few parameters and operations, it achieves fast inference and low power consumption—both of which are critical for running models on devices such as the Raspberry Pi 4 or Jetson Nano. By comparing its performance with that of deeper models, SAE-FF serves as a useful reference for analyzing the balance between computational efficiency and classification accuracy in edge-based network intelligence.

The classifier’s architecture is intentionally simple, featuring a single hidden layer composed of a 4-neuron input, a 4-neuron hidden layer, and a single-neuron output. To maintain learning speed and introduce non-linearity, the hidden layer uses the ReLU activation function. The output layer produces a logit score, and the BCEWithLogitsLoss() function handles the final binary classification.

This shallow structure results in a very small number of parameters and a low computational load (FLOPs), making it ideal for use on edge devices. The success of this shallow classifier hinges on the quality of the features supplied by the SAE; because the SAE provides a feature set with minimal information loss, the FF network can achieve high performance while remaining exceptionally lightweight.

#### 2.4.2. Deep Neural Network Classifier

The DNN classifier employed in this study takes the same 4-dimensional latent features as input, similar to the FF classifier, but adopts a deeper architecture to perform binary classification (see Figure 3). Compared with the FF classifier, this structure provides sufficient representational capacity while remaining lightweight enough to allow feasible inference on edge devices. To avoid excessive computational complexity, the architecture was restricted to a two-layer structure, thereby maintaining a balance between efficiency and performance for comparative evaluation.

The DNN consists of two fully connected layers, with the final output transformed into probabilities via a softmax function. For training, CrossEntropyLoss was employed as the objective function for binary classification. The SAE-DNN was chosen for its stronger representational capacity while maintaining manageable computational costs. In contrast to convolutional or transformer-based architectures—which demand substantially more parameters and FLOPs—the DNN provides sufficient depth to capture non-linear feature interactions within the SAE’s latent space, making it well suited for edge environments. This design achieves high classification accuracy without exceeding the memory and latency limits of devices such as the Jetson Nano and Coral Dev Board.

As such, SAE-DNN represents the upper bound of achievable accuracy within the proposed efficiency-oriented framework, serving as a balanced counterpart to the lightweight SAE-FF model.

### 2.5. Experimental Setup

#### Dataset

This study conducted its simulations using the CIC-IDS2017 dataset from the Canadian Institute for Cybersecurity at the University of New Brunswick [36]. The data was collected within a real-world network environment over five days, from 9:00 AM on Monday, 3 July 2017, to 5:00 PM on Friday, 7 July 2017. The dataset includes various attacks, such as Brute Force FTP, Brute Force SSH, DoS, Heartbleed, Web Attacks, Infiltration, Botnet, and DDoS, which were executed during the morning and afternoon sessions from Tuesday to Friday. The data from Monday consists exclusively of benign traffic.

The CIC-IDS2017 dataset is distributed with both raw packet-level data in PCAP files and CSV files containing statistical flow information generated by the CICFlowMeter tool. The CSV files are organized into two directories: GeneratedLabelledFlows and MachineLearningCSV. For this study, we utilized the CSV files located in the GeneratedLabelledFlows directory, with the details summarized in Table 3 and Table 4.

To mitigate issues arising from this imbalance, this paper excluded three CSV files in which the proportion of benign labels was greater than 98%: Thursday-WorkingHours-Morning-WebAttacks, Thursday-WorkingHours-Afternoon-Infilteration, and Friday-WorkingHours-Morning. This step was taken to prevent severe training bias and avoid distortions in the binary classification results.

### 2.6. Preprocessing

#### 2.6.1. Sampling

As described in the dataset overview, the distribution of attack types and the proportion of BENIGN labels differ considerably across the daily files. To address this variation and prepare a dataset suitable for training and evaluation, a multi-step preprocessing and sampling process was carried out. Because each raw PCAP file is over 5 GB in size, sampling was applied to reduce the processing load and to create a manageable experimental environment. The files used for preprocessing are listed in Table 5.

To avoid creating an unrealistically balanced dataset, we kept the ratio of normal to attack traffic at about 65:35 (normal: attack), which is closer to what is usually observed in enterprise or backbone networks where benign flows are dominant. Earlier intrusion detection datasets such as CIC-IDS2017, UNSW-NB15, and Bot-IoT also show benign traffic shares of roughly 60–80%, which supports this choice as a reasonable and reproducible baseline. This mild imbalance keeps the dataset realistic but still ensures that there are enough attack samples for stable training. We also used stratified 5-fold cross-validation to keep this ratio consistent across all folds.

After cleaning the selected samples (removing missing values) listed in Table 6, data were sampled separately for normal and abnormal classes. For the CSV files, 20% of the data were used, and for the PCAP files, 10% were selected. This choice was made because the PCAP files are much larger and require more computation and memory to process, while the CSV files contain ground-truth labels and were therefore assigned a higher sampling rate. Table 7 summarizes the final label distribution after sampling: 197,198 records from the CSV data and 3,533,869 from the PCAP data.

#### 2.6.2. Packet and Flow Features

To distinguish between normal and abnormal traffic, this study used a combination of 8 packet-level fields from PCAP files and 5 flow-level statistics from CSV files.

The first set of features was derived from the headers of individual packets. The 8 selected packet fields include packet length, protocol, TTL, source/destination port, source/destination IP address, and payload length. Other header fields were excluded because they contained fixed values (e.g., IP version, Header length) or were deemed uninformative for this classification task (e.g., Type of Service, Identification).

The second feature set describes the characteristics of each traffic flow. A flow is defined by a unique 5-tuple (source/destination IP, source/destination Port, protocol), and packets with the same 5-tuple that are close in time are grouped into a single flow. A total of 5 features were used to characterize each flow:Flow count: The number of packets belonging to the flow.Duration: The total time elapsed from the first to the last packet in the flow.Average packet length: The mean size of the packets within the flow.Total payload: The sum of the data payload sizes for all packets in the flow.Average TTL: The average Time-to-Live value, which can indicate the proximity of the network path.

These flow-based features were used to represent the transmission patterns and overall state of the traffic. A ground-truth label was assigned to each extracted feature set by creating a flow ID from the 5-tuple (src IP, dst IP, src port, dst port, protocol) and matching it with the corresponding labeled flow in the CSV files. For the binary classification task, labels were normalized so that ‘BENIGN’ was mapped to 0 (normal) and all attack types were mapped to 1 (abnormal).

While the initial CSV sampling yielded 197,198 flow-based records (as described in Section 2.6.1), the process of matching these flows to the extracted packet-level features resulted in a final dataset of 421,974 samples. This final dataset consisted of 286,876 normal samples and 135,098 abnormal samples. The reason why the number of sampled packet-level instances exceeds the number of sampled CSV flows (197,198) is that a single flow ID corresponds to multiple packets.

#### 2.6.3. Cross-Validation

To validate the models’ performance, we employed a flow-aware 5-fold cross-validation strategy. The dataset was first grouped by Flow ID to ensure that packets belonging to the same flow were not split across different folds, thus avoiding data leakage. These flow groups were then distributed across the five folds using a stratified approach that preserved the overall class ratio (≈65:35, normal:attack).

To prevent bias from disproportionately large normal flows, any single flow containing more than 50,000 packets was excluded from the fold creation process. Each fold contained approximately 63,399 samples, with a balanced class distribution of 58.4% Normal (36,379) and 42.6% Attack (27,020) records. This setup mitigated class imbalance issues between folds and ensured consistent evaluation across training and validation runs.

### 2.7. Implementation Environment

The experiments in this paper were conducted on a Linux server running the Ubuntu 20.04.6 LTS operating system. The system was equipped with a 12th-generation Intel i9-12900 processor (16 cores, 24 threads), 62 GB of memory, and an NVIDIA GeForce RTX 3080 GPU with 12 GB of VRAM. The GPU driver version was NVIDIA-SMI 570.133.07, and all experiments were implemented using the PyTorch framework. Real-time inference tests for the SAE-FF model were measured using the CPU. The key libraries used for the experiments included Python 3.8, PyTorch 2.4.1 (CPU-only build), and Scapy 2.4.3.

To evaluate the performance of the proposed SAE-FF and SAE-DNN lightweight classifiers in a realistic setting, inference tests were conducted on three edge devices: the Raspberry Pi 4 [40], Jetson Nano [41], and Coral Dev Board [42]. The specifications for each device are detailed in Table 8. For the Raspberry Pi 4, inference latency and power metrics were directly benchmarked, whereas for the Jetson Nano and Coral Dev Board, the results were partly modeled using official manufacturer specifications and DVFS/IPC-based estimation, due to limited hardware accessibility.

All models were trained using the Adam optimizer with a learning rate of 1 × 10^−3^ and a batch size of 1024. The number of training epochs was set to 30 for the SAE, 20 for the FF classifier, and 30 for the DNN classifier. A fixed random seed (42) was applied in all experiments to ensure reproducibility.

The sparse autoencoder employed a hidden dimension of 4, a sparsity objective (ρ = 0.05), and a penalty factor (β = 1 × 10^−3^). Both the SAE-FF and SAE-DNN models used the ReLU activation function, while the DNN additionally applied dropout (*p* = 0.2) for regularization. All experiments were conducted using 5-fold cross-validation under identical preprocessing and parameter configurations.

### 2.8. Performance Evaluation Method

The models’ classification performance was evaluated using standard metrics: Accuracy, Precision, Recall, F1-score, and AUC (Area Under the Curve). The equations used to calculate these metrics are defined in Table 9.

Accuracy measures the proportion of correctly classified samples out of the total samples.Precision measures the proportion of true positive samples among all samples classified as positive.Recall (or Sensitivity) measures the proportion of true positive samples that were correctly identified.The F1-score is the harmonic mean of Precision and Recall, providing a measure of a model’s balanced performance.AUC represents the area under the ROC curve and provides a comprehensive evaluation of a model’s performance across all classification thresholds. The F1-score and AUC were used to assess the stability and overall effectiveness of the models.

While the metrics in Table 9 assess the model’s classification accuracy, a practical analysis for edge environments must also consider computational efficiency. Therefore, we measure the model’s complexity (FLOPs) and inference time, which are used to calculate the comprehensive efficiency metric, EPES.

FLOPs [43] represent the total number of floating-point computations required for a single input and are widely used as an indicator of model complexity and inference workload. In neural networks, FLOPs are typically calculated for fully connected layers as FLOPs = 2 × input size × output size. A lower FLOPs value indicates a more lightweight and computationally efficient model. In this study, FLOPs were computed on the basis of a single-sample inference, where the number of multiply–accumulate (MAC) operations in a fully connected layer of size n × m was converted using the relation 1 MAC = 2 FLOPs (including bias and activation operations). To measure baseline inference time, five models generated through 5-fold cross-validation were evaluated. For each model, inference on the test dataset was repeated five times, and the average was calculated. The final baseline time was then obtained as the mean across the five folds. This multi-trial averaging minimized measurement error due to transient system load, thereby providing stable estimates. The derived baseline time was subsequently used to estimate inference latency on each edge device.

The baseline CPU used for inference measurement was the Intel i7-7700K (4.2 GHz), which is widely adopted as a reference processor for single-thread performance in various benchmarking studies. Inference latency on edge devices was estimated relative to the inference time measured on the Intel i7-7700K. Following the CPU-DVFS model, device-specific inference times were derived by applying both clock frequency scaling and architecture correction factors. Since performance differences cannot be fully explained by clock frequency alone, instruction-per-cycle (IPC)-based architectural correction factors (see Table 10) were incorporated to reduce estimation errors. For the Coral Dev Board, TPU acceleration was also considered, reducing inference time by a factor of 1/15 [42,43,44,45,46,47].

## 3. Result

### 3.1. Classification Performance Comparison

In this study, the performance of the SAE-FF was compared with that of the multi-layer perceptron-based SAE-DNN. Both models employed the same input feature vectors and preprocessing procedures. The main training parameters were set as follows: hidden_dim = 4, sparsity = 0.05, β = 1 × 10^−3^, and LR = 1 × 10^−3^. For sparsity, values of 0.03, 0.04, and 0.05 were tested, and the highest accuracy and most stable performance were achieved at 0.05.

Table 11 presents the 5-fold cross-validation results. SAE-FF achieved an average accuracy of 93.66% (±0.0103), whereas SAE-DNN attained 99.33% (±0.0005). The extremely low standard deviation of the DNN shows its high stability. Although SAE-DNN improved accuracy by ≈5.8% over SAE-FF, this gain required a substantially higher computational cost. The SAE-FF executed only ~40 FLOPs compared with ~8 960 FLOPs for SAE-DNN, demonstrating far greater efficiency. This trade-off between accuracy and cost implies that model selection should depend on the deployment environment, which also relates directly to the core rationale behind the proposed EPES metric.

All experiments were conducted with five-fold cross-validation to ensure reliability. The SAE encoder reduced each 13-dimensional input into a 4-dimensional latent vector, supporting model lightweighting. As shown in Table 11, both classifiers maintained stable F1-scores across all folds, with standard deviations below 0.011, demonstrating consistent learning behavior. The SAE-DNN achieved almost perfect AUC values (>0.999) in every fold, which indicates that its deeper hidden layers enhanced the discriminative capability of features extracted from the SAE latent space. This suggests that the stronger representational power of the SAE-DNN played a key role in classifying the non-linear boundaries that exist within the 4-dimensional latent space. In contrast, the single-layer SAE-FF performed well for linearly separable patterns but had difficulty learning more complex boundaries.

The SAE-FF showed slightly larger variations across folds (AUC = 0.966–0.988), indicating that one hidden layer was sufficient to capture the dominant traffic patterns but not sensitive enough to the minor variations among attack types. Section 3.2 further discusses how the theoretical computational gap (FLOPs) identified in this section translated into impacts in inference time, power consumption, and the overall EPES performance on real-world edge devices.

### 3.2. Performance Evaluation by Edge Device

The two classifiers were next evaluated using modeled benchmarks of three representative edge devices—Raspberry Pi 4, Jetson Nano, and Coral Dev Board—whose benchmark specifications are listed in Table 12. Accuracy scores from cross-validation were uniformly applied to all devices.

Device performance factors, measured relative to an Intel i7-7700K baseline, ranked as follows: Jetson Nano (0.2213) > Raspberry Pi 4 (0.2143) > Coral Dev Board (0.1964). In FLOPs efficiency, SAE-FF required 224 times fewer operations (40 vs. 8960), achieving an efficiency score of 0.9960. Inference latency was consistently shorter for SAE-FF because of its two-layer structure. On the Coral Dev Board, Edge TPU acceleration enabled 0.039 ms inference for SAE-FF and 0.770 ms for SAE-DNN. On the Raspberry Pi 4, the gap widened to ~19.7× (0.537 ms vs. 10.590 ms). Model sizes were ≈0.1 KB for SAE-FF and ≈18.6 KB for SAE-DNN.

Figure 4, Figure 5, Figure 6, Figure 7, Figure 8, Figure 9 and Figure 10 graphically compare the performance of SAE-FF and SAE-DNN across all edge devices. Figure 4 shows the overall accuracy distribution, with SAE-DNN consistently outperforming SAE-FF across all devices and maintaining nearly identical scores across folds.

Overall, both models achieved real-time inference, with SAE-FF favored for low-power devices and SAE-DNN effective on hardware-accelerated platforms. Figure 10 presents ROC-curve results: AUC = 0.9758 for SAE-FF and 0.9993 for SAE-DNN. The EPES decomposition in Table 12 further clarifies device-level differences. On the Jetson Nano, despite slightly higher power usage, its balanced CPU architecture achieved the top overall EPES (0.8370). The Raspberry Pi 4, although marginally slower, provided the most energy-efficient operation per watt. The Coral Dev Board exhibited the lowest latency owing to Edge TPU acceleration but showed reduced CPU performance, which constrained EPES gains.

Notably, the most critical finding is that the SAE-FF, despite being approximately 5.8% less accurate, recorded overwhelmingly higher total EPES on all devices than the SAE-DNN (e.g., 0.8359 vs. 0.6578 on the Raspberry Pi 4). This finding confirms the central premise of our study: that a single metric like accuracy is insufficient to evaluate a model’s utility in an edge environment. The decisive factor in this score disparity was the FLOPs efficiency score (0.9960 vs. 0.1040), indicating that in resource-constrained settings, computational efficiency is a crucial factor, important enough to justify the trade-off of a moderate loss in accuracy.

Figure 5 shows the inference latency results, with SAE-FF running significantly faster on a Raspberry Pi 4 with up to 19.7× lower latency, while SAE-DNN achieves near-real-time performance only on the Coral Dev Board due to edge TPU acceleration. Figure 6, Figure 7, Figure 8 and Figure 9 illustrate the scalability and efficiency characteristics of the two models. In particular, Figure 6 shows that inference time scales almost linearly with FLOPs, and Figure 7 compares energy efficiency across devices, demonstrating that SAE-FF offers superior performance per watt. Figure 8 demonstrates that FLOP efficiency is a key factor influencing overall EPES. Figure 9 presents a comprehensive comparison of device-level EPES values by integrating multiple performance components.

Finally, Figure 10 shows the ROC curves for the two classifiers. Both models outperform random guessing (AUC = 0.5). The SAE-DNN curve closely follows the upper left boundary, demonstrating very high recall and precision.

## 4. Discussion

The results clearly show a trade-off between classification accuracy and computational efficiency in the SAE-based classifiers. The deeper architecture of SAE-DNN improves the model’s ability to represent complex traffic behaviors, resulting in nearly perfect classification performance. This improvement comes at the cost of a roughly 225-fold increase in FLOPs. By contrast, SAE-FF achieves acceptable accuracy with far lower computational demand and model size—an essential property for energy-constrained edge environments.

From an architectural standpoint, this trade-off can be explained by the difference in representational depth. The SAE-DNN benefits from multiple hidden layers that can learn highly non-linear relationships in the latent feature space generated by the autoencoder. Meanwhile, the SAE-FF relies on a single fully connected layer that captures only the dominant traffic patterns. These tendencies align with findings in prior studies on lightweight models like MobileNets and EfficientNet, which also indicate that increasing model depth enhances generalization but raises computational costs, particularly in embedded settings. This study empirically demonstrates that this well-known trade-off is a critical consideration not only in computer vision but also in the domain of network anomaly detection.

The hardware experiments further highlight the importance of system-level constraints. On the Raspberry Pi 4, SAE-FF achieved real-time inference without any accelerator support, whereas SAE-DNN required TPU assistance on the Coral Dev Board to obtain comparable latency. The Jetson Nano exhibited balanced throughput and power consumption, suggesting it as a suitable middle-ground platform for medium-scale deployment. These observations emphasize that model selection at the edge should be guided not only by accuracy but also by the available resources—latency tolerance, power budget, and memory footprint.

The proposed EPES offers a comprehensive way to quantify this balance. By integrating accuracy, latency, FLOPs, and power consumption into a single interpretable metric, EPES enables fair comparison across heterogeneous devices and models. Unlike conventional evaluations that emphasize a single indicator such as accuracy, EPES reveals an “efficiency frontier”. Our results clearly illustrate this frontier: improving accuracy by approximately 5.8% (from 93.66% to 99.33%) required a 225-fold increase in computational cost. EPES precisely captures this inflection point where marginal gains in accuracy demand an exponential cost in efficiency. This provides a practical tool for choosing the most suitable model when deploying deep learning systems in diverse edge environments.

Beyond quantitative evaluation, EPES also carries practical implications for adaptive edge intelligence. For instance, it can enable context-aware inference systems that dynamically select models based on real-time conditions. An edge device could run the high-accuracy SAE-DNN when connected to a main power source but autonomously switch to the power-efficient SAE-FF when operating on battery. EPES could serve as the core threshold for making such dynamic decisions, significantly enhancing the system’s autonomy and operational longevity.

This flexibility directly contributes to the design of scalable and energy-efficient AI systems, particularly in the IoT and SDN domains. In an SDN environment, a central controller can monitor the resource status of each edge node (e.g., switches) in real-time. Based on each node’s available resources, the controller could use EPES to dynamically deploy the most appropriate lightweight anomaly detection model. This allows for an intelligent traffic management policy that optimizes both security and efficiency across the entire network.

Overall, these findings suggest that effective anomaly detection at the edge requires a holistic perspective that integrates both algorithmic performance and hardware constraints. Through the EPES framework, SAE-based lightweight models strike a balanced compromise between accuracy and efficiency, offering a strong foundation for the next generation of intelligent and sustainable edge–AI systems.

## 5. Conclusions

This study proposed a comparative framework for lightweight anomaly-traffic classification that combines Sparse Autoencoder-based feature extraction with two classifiers—an FF network and a DNN. Using the CIC-IDS2017 dataset and five-fold cross-validation, our findings revealed that while the SAE-DNN achieved the highest accuracy, the SAE-FF offered far superior computational efficiency and, consequently, the best overall EPES.

The introduced EPES metric provides a unified benchmark that integrates accuracy, latency, power consumption, and FLOPs, enabling a balanced evaluation of edge–AI models. The overall framework contributes to edge-intelligence research by bridging algorithmic design and hardware constraints, offering quantitative guidance for model deployment on heterogeneous devices.

Nevertheless, this study has limitations that present clear opportunities for future research. The analysis relied on a single dataset (CIC-IDS2017), assumed binary classification, and partly used estimated latency and power metrics.

Future work will address these issues by validating the framework on broader datasets (e.g., UNSW-NB15, Bot-IoT), exploring federated and continual learning, and applying model compression with hardware-specific optimization (e.g., TensorRT, TFLite). Through these extensions, this line of research aims to contribute to the development of more practical and efficient AI-driven security solutions for resource-constrained networks.

## Figures and Tables

**Figure 1 sensors-25-06439-f001:**
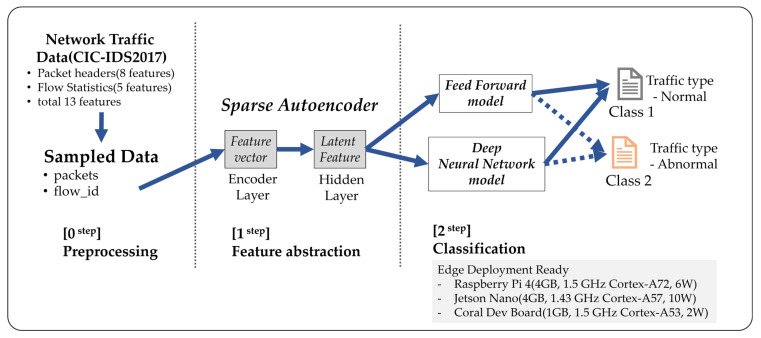
Overview of the proposed system architecture.

**Figure 2 sensors-25-06439-f002:**
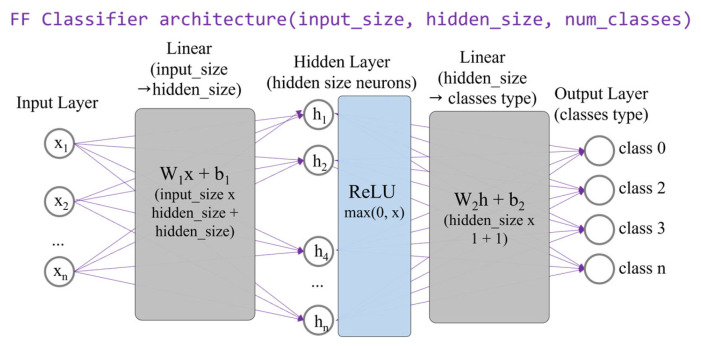
Feed-forward classifier architecture.

**Figure 3 sensors-25-06439-f003:**
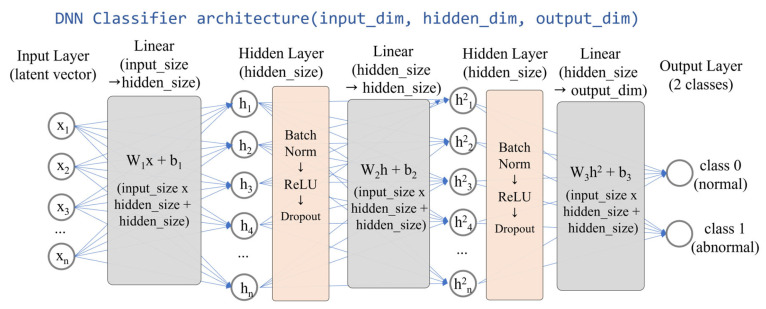
Deep neural network classifier architecture.

**Figure 4 sensors-25-06439-f004:**
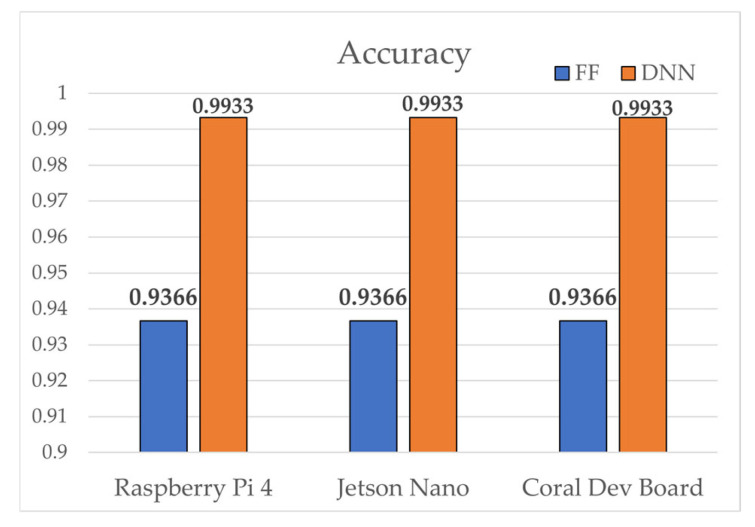
Comparison of 5-fold cross-validation performance (accuracy) for SAE-FF and SAE-DNN.

**Figure 5 sensors-25-06439-f005:**
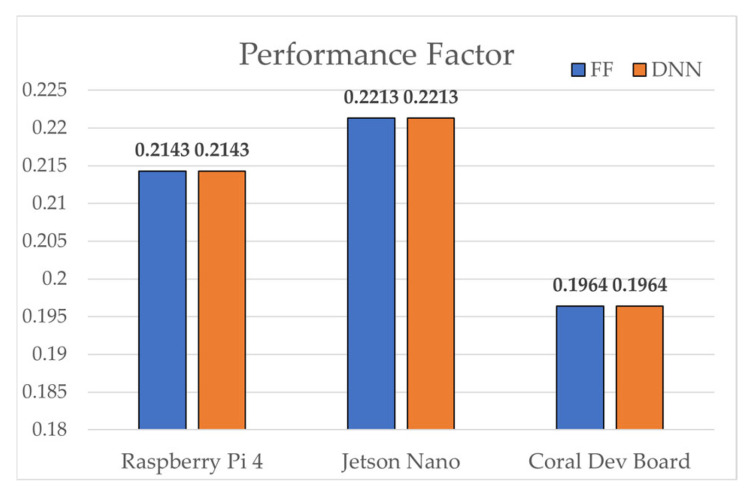
Comparison of performance factor across edge devices.

**Figure 6 sensors-25-06439-f006:**
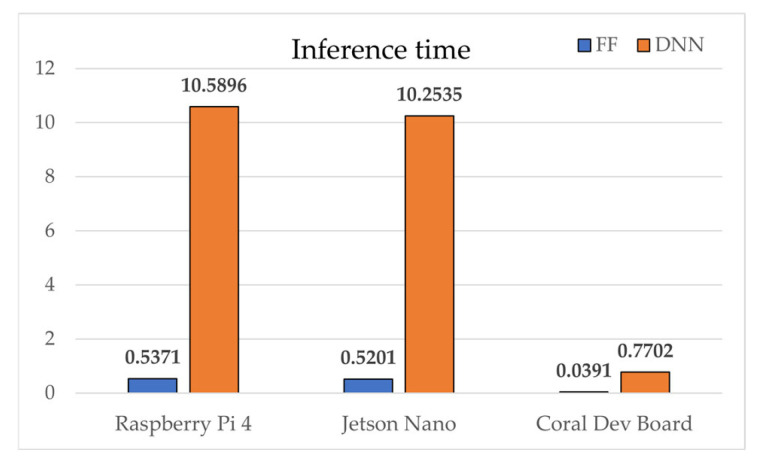
Comparison of inference time across edge devices.

**Figure 7 sensors-25-06439-f007:**
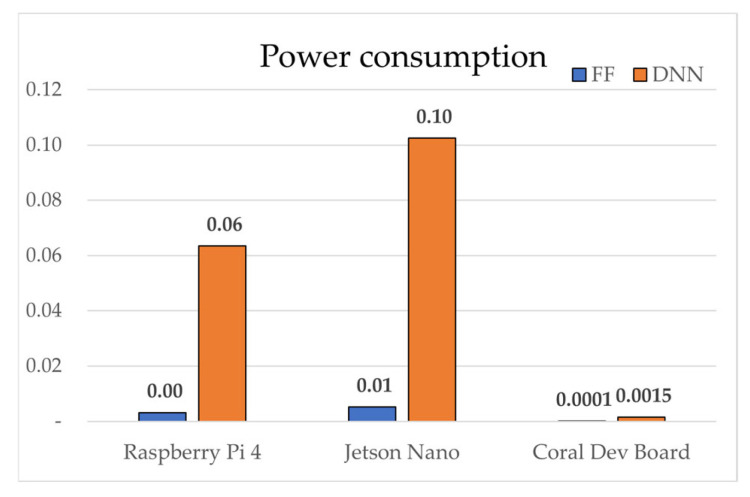
Comparison of power consumption across edge devices.

**Figure 8 sensors-25-06439-f008:**
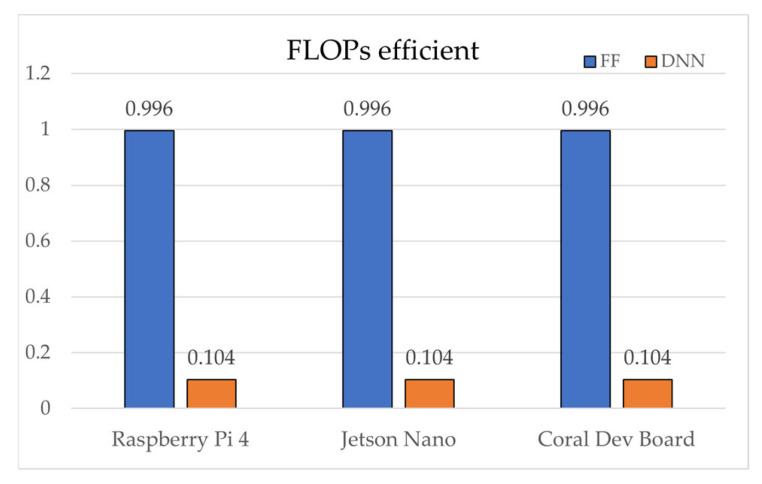
Comparison of FLOPs efficiency across edge devices.

**Figure 9 sensors-25-06439-f009:**
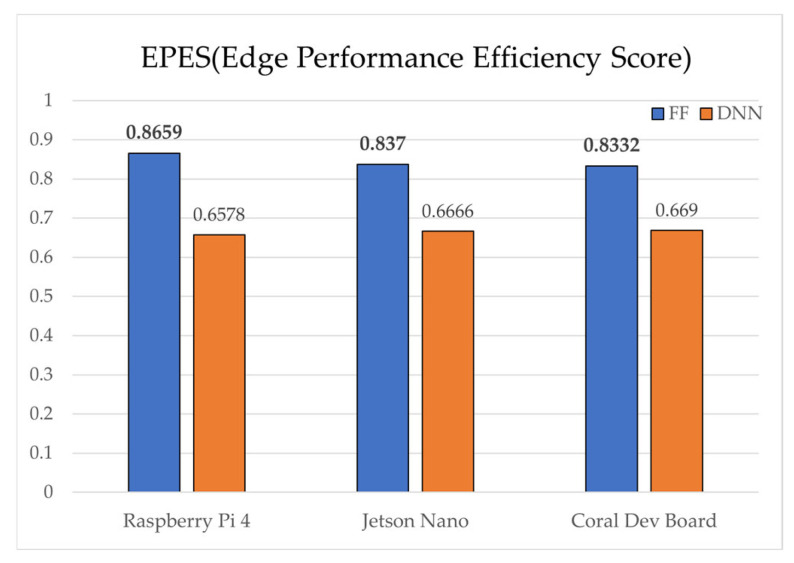
Comparison of total EPES for SAE-FF and SAE-DNN.

**Figure 10 sensors-25-06439-f010:**
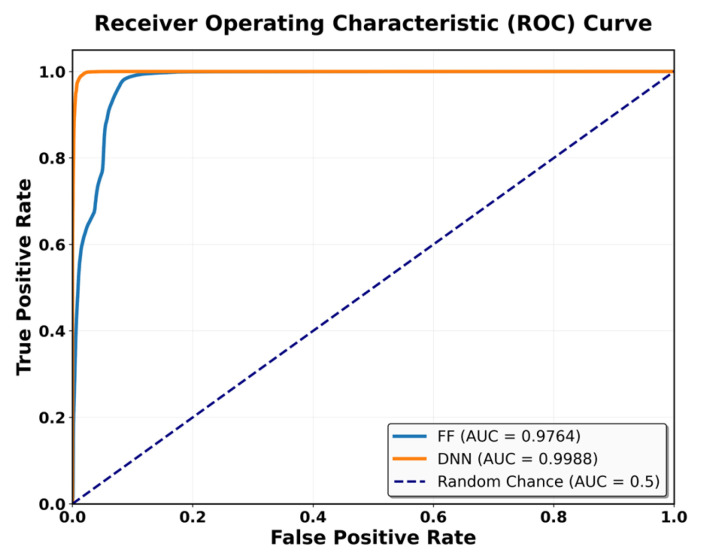
Comparison of ROC curve performance between the SAE-FF and SAE-DNN models.

**Table 1 sensors-25-06439-t001:** Comparison of evaluation scope with related studies (√ = included, ○ = partially reported, - = not reported).

Study	Accuracy	F1/AUC	Inference Latency	FLOPs	Memory	Power (Measured)	Device-Level Deployment Suitability	EPES
Sparse Autoencoder–based feature extraction [23,24,25]	√	○	√	-	-	-	-	-
Lightweight deep learning [8,31,33,34]	√	√	-	-	-	○	-	-
Mobile/lightweight application studies	√	√	√	√	-	-	-	-
This Work	√	√	√	√	√	√ (with modeling)	√ (RasPi/Jetson/Coral)	√ (newly proposed)

**Table 2 sensors-25-06439-t002:** Calculation methods, normalization procedures, and weighting factors of EPES components.

Component		Calculation Method	Normalization	Weight
EPES	Speed Efficiency	Inference time evaluated against a 10 ms baseline	Min (1.0, 10.0/time_ms)	25%
	Accuracy	Average accuracy from 5-fold validation	Direct accuracy value	25%
	Memory Efficiency	Model size (MB) + Data size (MB)	Max (0, (512 − total_memory)/512)	15%
	FLOPs Efficiency	2 × Inputs × Output	Max (0, (10,000 − FLOPs)/10,000))	20%
	Performance Factor (cpu_score)	(device_CPU (GHz)/4.2 (GHz)) × arch_factor	Direct performance score	15%
Reference	Energy consumption	Device power (W) × Inference time (s)	-	-

**Table 3 sensors-25-06439-t003:** CIC-IDS2017 CSV file information.

CSV File List		Flows	Columns
Tuesday	Tuesday-WorkingHours.pcap_ISCX.csv	445,909	85
Wednesday	Wednesday-workingHours.pcap_ISCX.csv	692,703	
Thursday	Thursday-WorkingHours-Morning-WebAttacks.pcap_ISCX.csv	170,335	
	Thursday-WorkingHours-Afternoon-Infilteration.pcap_ISCX.csv	288,602	
Friday	Friday-WorkingHours-Morning.pcap_ISCX.csv	191,033	
	Friday-WorkingHours-Afternoon-DDos.pcap_ISCX.csv	225,745	
	Friday-WorkingHours-Afternoon-PortScan.pcap_ISCX.csv	286,467	

**Table 4 sensors-25-06439-t004:** Traffic Distribution by Attack Type and Day (CIC-IDS2017 Dataset).

Type	Day						
	Tues	Wednes	Thurs-M	Thrus-A	Fri-M	Fri-A-DDos	Fri-A-PortScan
BENIGN	432,074 (96.9%)	440,031 (63.5%)	168,186 (98.7%)	288,566 (100%)	189,067 (99.0%)	97,718 (43.3%)	127,537 (44.5%)
FTP-Patator	7938 (1.8%)	0 (0.0%)	0 (0.0%)	0 (0.0%)	0 (0.0%)	0 (0.0%)	0 (0.0%)
SSH-Patator	5897 (1.3%)	0 (0.0%)	0 (0.0%)	0 (0.0%)	0 (0.0%)	0 (0.0%)	0 (0.0%)
DoS Hulk	0 (0.0%)	231,073 (33.4%)	0 (0.0%)	0 (0.0%)	0 (0.0%)	0 (0.0%)	0 (0.0%)
DoS GoldenEye	0 (0.0%)	10,293 (1.5%)	0 (0.0%)	0 (0.0%)	0 (0.0%)	0 (0.0%)	0 (0.0%)
DoS slowloris	0 (0.0%)	5796 (0.8%)	0 (0.0%)	0 (0.0%)	0 (0.0%)	0 (0.0%)	0 (0.0%)
DoS Slowhttptest	0 (0.0%)	5499 (0.8%)	0 (0.0%)	0 (0.0%)	0 (0.0%)	0 (0.0%)	0 (0.0%)
Heartbleed	0 (0.0%)	11 (0.0%)	0 (0.0%)	0 (0.0%)	0 (0.0%)	0 (0.0%)	0 (0.0%)
Web Attack Brute Force	0 (0.0%)	0 (0.0%)	1507 (0.9%)	0 (0.0%)	0 (0.0%)	0 (0.0%)	0 (0.0%)
Web Attack XSS	0 (0.0%)	0 (0.0%)	652 (0.4%)	0 (0.0%)	0 (0.0%)	0 (0.0%)	0 (0.0%)
Web Attack Sql Injection	0 (0.0%)	0 (0.0%)	21 (0.0%)	0 (0.0%)	0 (0.0%)	0 (0.0%)	0 (0.0%)
Infiltration	0 (0.0%)	0 (0.0%)	0 (0.0%)	36 (0.0%)	0 (0.0%)	0 (0.0%)	0 (0.0%)
Bot	0 (0.0%)	0 (0.0%)	0 (0.0%)	0 (0.0%)	1966 (1.0%)	0 (0.0%)	0 (0.0%)
DDoS	0 (0.0%)	0 (0.0%)	0 (0.0%)	0 (0.0%)	0 (0.0%)	128,027(56.7%)	0 (0.0%)
PortScan	0 (0.0%)	0 (0.0%)	0 (0.0%)	0 (0.0%)	0 (0.0%)	0 (0.0%)	158,930(55.5%)
Total	445,909	692,703	170,366	288,602	191,033	225,745	286,467

**Table 5 sensors-25-06439-t005:** Preprocessed file information.

Filename	Normal	Abnormal	CSV File Size (GB)	PCAP File Size (GB)
Tuesday-WorkingHours	432,074 (96.9%)	13,835 (3.1%)	0.1627	10.2895
Wednesday-workingHours	440,031 (63.5%)	252,672 (36.5%)	0.2660	12.4990
Friday-WorkingHours-Afternoon-DDos	97,718 (43.3%)	128,027 (56.7%)	0.0895	8.2322
Friday-WorkingHours-Afternoon-PortScan	127,537 (44.5%)	158,930 (55.5%)	0.0948	
Total: 1,650,824 (100%)	1,097,360 (66.47%)	553,464 (33.53%)		

**Table 6 sensors-25-06439-t006:** CSV sampled file summary (based on a maximum of 60,000 records per file).

Filename	Original	Sampled (Ratio (%))
Tuesday-WorkingHours	445,708	56,455 (12.7)
Wednesday-workingHours	691,695	60,000 (8.7)
Friday-WorkingHours-Afternoon-DDos	286,452	39,173 (13.7)
Friday-WorkingHours-Afternoon-PortScan	225,741	41,570 (18.4)
Total	1,649,596	197,198 (11.95)

**Table 7 sensors-25-06439-t007:** Label distribution of sampled data by type (counts and percentages).

Type	Flows (Ratio (%))
BENIGN	121,846 (61.8)
PortScan	20,224 (10.3)
DDoS	18,999 (9.6)
DoS Hulk	15,559 (7.9)
FTP-Patator	6012 (3.0)
SSH-Patator	5897 (3.0)
DoS GoldenEye	5728 (2.9)
DoS slowloris	2456 (1.2)
DoS Slowhttptest	471 (0.2)
Heartbleed	6 (0.0)
Total	197,198 (100)

**Table 8 sensors-25-06439-t008:** Detailed Specifications of Each Edge Device.

Device	Raspberry Pi 4 (Actual Benchmark Specs)	Jetson Nano (Official NVIDIA Specs)	Coral Dev Board (Official Google Specs)
Memory	4 GB	4 GB	1 GB
CPU	Quad-A72 1.5 GHz	Quad-A57 1.43 GHz	Quad-A53 1.5 GHz
Power	Approx. 6–10 W	Max 10 W	Edge TPU: 2 W

**Table 9 sensors-25-06439-t009:** Performance Metric Definitions.

Metric	Equation
Accuracy	(TP + TN)/(TP + TN + FP + FN)
Precision	TP/(TP + FP)
Recall	TP/(TP + FN)
F1-Score	2 × (Precision × Recall)/(Precision + Recall)

**Table 10 sensors-25-06439-t010:** CPU and Architecture Correction Factor for Each Device.

Device	Raspberry Pi 4 (Actual Benchmark Specs)	Jetson Nano (Official NVIDIA Specs)	Coral Dev Board (Official Google Specs)
CPU	Quad-A72 1.5 GHz	Quad-A57 1.43 GHz	Quad-A53 1.5 GHz
Architecture Correction Factor [43]	0.6	0.65	0.55
Baseline CPU: Intel i7-7700K (4.2 GHz)			

**Table 11 sensors-25-06439-t011:** 5-fold cross-validation results for performance comparison of SAE-FF and SAE-DNN.

Fold	SAE-Feed Forward			SAE-Deep Neural Network		
	Accuracy	F1-Score	AUC	Accuracy	F1-Score	AUC
1/5	0.9238	0.9242	0.9664	0.9932	0.9933	0.9990
2/5	0.9539	0.9541	0.9906	0.9936	0.9936	0.9994
3/5	0.9306	0.931	0.9685	0.9924	0.9924	0.9991
4/5	0.9332	0.9335	0.9750	0.9935	0.9935	0.9994
5/5	0.9413	0.9416	0.9885	0.9938	0.9938	0.9994
Average	0.9366 ± 0.0103	0.9369 ± 0.0103	0.9758 ± 0.0086	0.9933 ± 0.0005	0.9933 ± 0.0005	0.9993 ± 0.0002

**Table 12 sensors-25-06439-t012:** Device-specific evaluation factors and total EPES comparison for SAE-FF/SAE-DNN.

Metric		SAE-FF			SAE-DNN		
		Raspberry Pi 4	Jetson Nano	Coral Dev Board	Raspberry Pi 4	Jetson Nano	Coral Dev Board
Inference Time (ms)		0.5371	0.5201	0.0391	10.5896	10.2535	0.7702
Memory Usage (MB)		100.97	100.97	100.97	100.99	100.99	100.99
Model Size (MB)		0.0001	0.0001	0.0001	0.0186	0.0186	0.0186
Power Consumption (W)		0.0032	0.0052	0.0001	0.0635	0.1025	0.0015
FLOPs		40	40	40	8960	8960	8960
EPES	Accuracy	0.9366	0.9366	0.9366	0.9933	0.9933	0.9933
	Performance Factor	0.2143	0.2213	0.1964	0.2143	0.2213	0.1964
	Speed Efficiency	1.0000	1.0000	1.0000	0.9443	0.9753	1.0000
	Memory Efficiency	0.8028	0.8028	0.8028	0.8028	0.8028	0.8028
	FLOPs Efficiency	0.9960	0.9960	0.9960	0.1040	0.1040	0.1040
	Total	0.8359	0.8370	0.8332	0.6578	0.6666	0.6690

## Data Availability

The data supporting the findings of this study are publicly available in the CIC-IDS2017 dataset, provided by the Canadian Institute for Cybersecurity at the University of New Brunswick, at https://www.unb.ca/cic/datasets/ids-2017.html, accessed on 15 October 2025. No new data were created or analyzed in this study.

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
