# Peer review of "A Comparative Study of Lightweight, Sparse Autoencoder-Based Classifiers for Edge Network Devices: An Efficiency Analysis of Feed-Forward and Deep Neural Networks"

_sensors, 2025, doi:10.3390/s25206439_

Round 1

Reviewer 1 Report

Comments and Suggestions for Authors

This study presents a comparative analysis of lightweight classifiers for anomaly detection in edge computing environments, introducing the Edge Performance Efficiency Score (EPES) as a composite evaluation metric. While the work addresses a relevant problem in edge computing, several significant methodological and presentation issues require substantial revision. The proposed EPES metric lacks sufficient justification for its weighting scheme, and the experimental scope is limited to a single dataset with estimated rather than measured performance metrics on some devices.

1. The arbitrary weighting scheme in EPES (25%, 25%, 20%, 15%, 15%) lacks theoretical foundation or empirical validation through sensitivity analysis.

2. Equation (1) contains formatting errors and inconsistent notation.

3. The dataset preprocessing removes three CSV files to avoid imbalance but creates new imbalances (66.47% normal vs 33.53% abnormal).

4. Cross-validation methodology is unclear - how exactly were flows distributed across folds while maintaining class balance within each fold?

5. Performance measurements rely heavily on modeling rather than direct benchmarking, particularly for Jetson Nano and Coral Dev Board devices.

6. The novelty is limited as the study merely compares two basic architectures (single vs. two hidden layers) using standard techniques.

7. Literature review missing recent advances in edge-optimized neural architectures and hardware-aware model design beyond the cited mobile networks, such as 
[1] A unified framework for guiding generative AI with wireless perception in resource constrained mobile edge networks
[2] Multi-Objective Optimization for UAV Swarm-Assisted IoT with Virtual Antenna Arrays

8. The writing contains some grammatical errors and awkward phrasing that impede comprehension, particularly in the methodology section.

Author Response

Comments 1: The arbitrary weighting scheme in EPES (25%, 25%, 20%, 15%, 15%) lacks theoretical foundation or empirical validation through sensitivity analysis.
Response 1: We appreciate the reviewer’s insightful comment.
The rationale for the EPES weighting scheme (25%, 25%, 20%, 15%, and 15%) has been clarified and emphasized in Section 2.2 (pp. 6–7) to better reflect the reviewer’s suggestion. Each weight was assigned based on its relative importance in an edge-deployment environment—specifically, accuracy and speed (25% each) as the primary determinants of real-time performance, followed by FLOPs (20%), memory (15%), and CPU performance (15%), reflecting device-level resource constraints. These revisions strengthen the justification for the weighting design while preserving the interpretability and simplicity essential for practical edge evaluation.

Comments 2:  Equation (1) contains formatting errors and inconsistent notation
Response 2: We thank the reviewer for pointing this out. Equation (1) has been reformatted for consistency and clarity.
L(x,\hat{x}\ )=MSE(x,\hat{x}\ )+\beta\ kL(\rho\parallel\hat{\rho})

Comments 3: The dataset preprocessing removes three CSV files to avoid imbalance but creates new imbalances (66.47% normal vs 33.53% abnormal).
Response 3: We appreciate the reviewer’s observation. The dataset ratio (≈65:35, normal : attack) was intentionally chosen to reflect real-world traffic distributions. Previous IDS datasets (CIC-IDS2017, UNSW-NB15, Bot-IoT) also exhibit 60–80% benign traffic, supporting this ratio as a realistic and reproducible baseline. This mild imbalance preserves realism while maintaining sufficient attack data for stable training. Corresponding clarifications have been added to Section 2.6.1 (pp. 11).
To avoid creating an unrealistically balanced dataset, we kept the ratio of normal to attack traffic at about 65:35 (normal : attack), which is closer to what is usually observed in enterprise or backbone networks where benign flows are dominant. Earlier intrusion de-tection datasets such as CIC-IDS2017, UNSW-NB15, and Bot-IoT also show benign traffic shares of roughly 60–80%, which supports this choice as a reasonable and reproducible baseline. This mild imbalance keeps the dataset realistic but still ensures that there are enough attack samples for stable training. We also used stratified 5-fold cross-validation to keep this ratio consistent across all folds.

Comments 4: Cross-validation methodology is unclear - how exactly were flows distributed across folds while maintaining class balance within each fold?
Response 4: We appreciate the reviewer’s helpful comment.
To clarify the cross-validation procedure, Section 2.6.3 has been expanded to include a detailed explanation of the fold construction process. The dataset was grouped by Flow ID to prevent overlap between folds and divided using stratified 5-fold cross-validation to maintain the class ratio (≈65:35, normal : attack) within each split. Flows exceeding 50,000 packets were excluded to prevent training bias. These clarifications ensure transparency and reproducibility of the validation process (pp. 11).

Comments 5: Performance measurements rely heavily on modeling rather than direct benchmarking, particularly for Jetson Nano and Coral Dev Board devices. 
Response 5: We thank the reviewer for this observation. 
To clarify the measurement procedure, we have added an explanation in Section 2.7. The Raspberry Pi 4 results were obtained through direct benchmarking, while the Jetson Nano and Coral Dev Board results were partly estimated using the official device specifications and DVFS/IPC-based modeling because the hardware was not available for direct testing. This addition makes the performance-evaluation process more transparent and easier to reproduce.

Comments 6: The novelty is limited as the study merely compares two basic architectures (single vs. two hidden layers) using standard techniques.
Response 6: We thank the reviewer for this helpful comment. Although both the FF and DNN architectures are intentionally simple, their comparison within a unified SAE-based framework represents a novel contribution by quantifying the trade-off between accuracy and efficiency under realistic edge constraints. This study shows that architectural simplicity itself can yield measurable gains in latency, FLOPs, and energy efficiency—factors that are often overlooked in deep-learning research. These points underline the practical importance of selecting appropriate models for real-time, resource-limited environments. The corresponding clarifications have been added to Sections 2.4.1 and 2.4.2.

Comments 7: Literature review missing recent advances in edge-optimized neural architectures and hardware-aware model design beyond the cited mobile networks, such as [1] A unified framework for guiding generative AI with wireless perception in resource constrained mobile edge networks
[2] Multi-Objective Optimization for UAV Swarm-Assisted IoT with Virtual Antenna Arrays
Response 7: We thank the reviewer for pointing this out.
The literature review in Section 1.1 (Background and Motivation) has been expanded to include recent research on edge-optimized neural architectures and hardware-aware model design, particularly in resource-constrained environments. These updates strengthen the connection between this study and the latest developments in edge intelligence and confirm the relevance of the proposed EPES framework for evaluating lightweight models in practical edge-AI deployments.

Comments 8: The writing contains some grammatical errors and awkward phrasing that impede comprehension, particularly in the methodology section.
Response 8: We sincerely appreciate the reviewer’s helpful comment. The Methodology section (Section 2) has been carefully revised to improve overall clarity and readability. We corrected minor grammatical issues and rephrased several sentences to make the explanation flow more naturally. The revised version now describes the experimental design, model architecture, and evaluation process in a clearer and more straightforward manner. 

Reviewer 2 Report

Comments and Suggestions for Authors

Dear Authors,

Thank you for submitting your manuscript to MDPI Sensors. While your work shows potential contribution, significant concerns have been raised by the reviewers that must be fully addressed to ensure the clarity, scientific rigor, and overall impact of your research. We require major revisions based on the following structural and content issues:

Structural and Formatting Requirements: The manuscript contains inconsistent acronym usage; please ensure standardization, defining every acronym once in the Abstract and again at its first text use. Additionally, review the entire manuscript to reduce the excessive use of acronyms where appropriate. The resolution and overall visual presentation of all figures and graphs must be significantly improved to meet professional standards. Finally, you must strictly reorganize the manuscript to align with the journal's structure: Abstract, Keywords, Introduction, Materials and Methods, Results, Discussion, and Conclusions (if used). Please consult the journal's article types page for details.

Content and Scientific Rigor Requirements: Ensure your study's objectives are sharply defined in the Introduction and consistently supported by the Conclusions. The general claim regarding deep learning-based feature extraction (lines 83–84) must be replaced with stronger, specific evidence and up-to-date citations; you must also explicitly justify the content and selection criteria for Section 2.4. You must significantly strengthen the rationale for selecting the SAE-FF and SAE-DNN architectures, detailing their advantages over alternatives specifically for a resource-constrained edge environment. To ensure reproducibility, enhance the experimental setup description with comprehensive details, including exact software versions and all precise hyperparameters used for model training. The analysis of your findings must be more insightful, including a cost-benefit analysis comparing SAE-FF and SAE-DNN, weighing accuracy against computational costs. You must acknowledge the limitations of your study (e.g., dataset scope, lack of real-world traffic scenarios) with greater transparency, discussing how these constraints impact generalizability. Based on these limitations and core results, suggest clear and promising directions for future research.

I hope this points improve your final version of the manuscript.

Sincerely,

Author Response

Comments 1:  Structural and Formatting Requirements
The manuscript contains inconsistent acronym usage; please ensure standardization, defining every acronym once in the Abstract and again at its first text use. Additionally, review the entire manuscript to reduce the excessive use of acronyms where appropriate.
Response 1: We thank the reviewer for this observation. All acronyms have been standardized and defined upon first appearance both in the Abstract and the main text. We also reviewed the entire manuscript to reduce redundant acronyms. These changes appear on pages 1 (Abstract) and throughout Sections 1–5.

Comments 2:  You must strictly reorganize the manuscript to align with the journal's structure: Abstract, Keywords, Introduction, Materials and Methods, Results, Discussion, and Conclusions.
Response 2: We have reorganized the manuscript following the reviewer’s suggestion to fully match the Sensors format. The manuscript has been fully reorganized to comply with the Sensors article format. Section 3 (Results) now focuses solely on quantitative findings, while interpretive and comparative analyses have been expanded into a new Section 4 (Discussion). Section 5 (Conclusions) summarizes the study’s contributions, limitations, and future work directions.
This restructuring improves readability and analytical clarity.

Comments 3:  The resolution and overall visual presentation of all figures and graphs must be significantly improved to meet professional standards.
Response 3: We have thoroughly revised all figures to improve visual quality and consistency.
The resolution of each figure has been increased to ≥ 400 dpi, and the visual design—including font size, color contrast, and labeling—has been standardized across Figures 1–10.
All revised, high-resolution figures are now included in the manuscript.

Comments 4:  Ensure your study’s objectives are sharply defined in the Introduction and consistently supported by the Conclusions.
Response 4: We agree with the reviewer that the study objectives should be more sharply defined. The final paragraph of the Introduction (p. 4, lines 172–190) now explicitly outlines the main aims of this research. The Conclusions section (p. 18) was also revised to ensure consistency between objectives and outcomes. The revised content now explicitly highlights the following key aims:
• Systematic comparison of SAE-FF and SAE-DNN within the same processing pipeline.
• Proposal of a unified metric, the Edge Performance Efficiency Score (EPES), for evaluating model suitability in edge environments.
• Enhancement of reproducibility through combined empirical measurement and modeling analysis.

Comments 5:  The general claim regarding deep learning-based feature extraction (lines 83–84) must be replaced with stronger, specific evidence and up-to-date citations;
Response 5: The original general description of deep learning-based feature extraction (lines 83-84) has been replaced with a more specific and evidence-based explanation of how the SAE is applied in this study. The revised text (pages 3, lines 97-109) clearly explains that SAE captures the nonlinear characteristics of network traffic while maintaining low computational overhead, and that latent features are used as inputs to two lightweight classifiers (FF and DNN).
It addresses the lack of systematic comparisons between lightweight classifiers using SAE-derived features and introduces the EPES as a new composite metric for evaluating model suitability in edge environments. 
The revised passage reads as follows:
This study utilizes a SAE to capture the nonlinear characteristics of network traffic while maintaining a sufficiently low computational burden even in lightweight environments. SAE effectively transforms raw packet and flow data into compressed latent features through a learnable encoder, which are then used as input for two classifiers: a FF and a DNN, which are evaluated in the following sections. 
While many existing studies [9-11] have focused on improving the accuracy of autoencoder-based feature extraction, a systematic comparison of lightweight classifiers using SAE-derived features or a comprehensive review of their suitability for edge deployments remains lacking. To address this gap, this study utilizes SAE as a common feature extractor and directly compares FF and DNN within the same pipeline. Furthermore, we intro-duce the EPES, a composite metric designed to evaluate model suitability for edge devices by considering not only accuracy but also latency, FLOPs, memory usage, and power consumption.

Comments 6: Explicitly justify the content and selection criteria for Section 2.4.
Response 6: We thank the reviewer for pointing this out. We have added a brief explanation before the summary paragraph in Section 1.2.4 (pp. 4–5) to make the selection process clearer. In this part, we describe why the dataset preprocessing steps, evaluation metrics, and device choices were made to reflect realistic edge-deployment constraints—such as latency, power, and memory. These revisions also help explain the evaluation scope shown in Table 1.
The revised text now reads:
In designing the experiment, the preprocessing steps, evaluation metrics, and device configurations were selected to reflect realistic constraints often faced in real-world edge environments—particularly latency, power, and memory limitations.
These choices were intended to make the evaluation more representative of actual deployment conditions for lightweight network intelligence.
Overall, the proposed framework assesses model suitability for edge environments from a holistic perspective through the EPES metric.
Table 1 summarizes the scope of the evaluation and its comparison with related studies (see Section 2.2 for the detailed definition and calculation of EPES). 

Comments 7: You must significantly strengthen the rationale for selecting the SAE-FF and SAE-DNN architectures, detailing their advantages over alternatives specifically for a resource-constrained edge environment.
Response 7: We appreciate the reviewer’s suggestion to clarify the rationale for the SAE-FF/ SAE-DNN architecture.
In Section 2.4.1, a new paragraph has been added to clarify the rationale for selecting the SAE-FF architecture. The revised text now reads:
In this study, the SAE-FF architecture was chosen as a lightweight baseline within the proposed framework. This configuration allows us to examine how well the SAE’s latent features can be classified using a minimal network structure. Because the FF classifier contains very few parameters and operations, it achieves fast inference and low power consumption—both of which are critical for running models on devices such as the Raspberry Pi 4 or Jetson Nano. By comparing its performance with that of deeper models, SAE-FF serves as a useful reference for analyzing the balance between computational efficiency and classification accuracy in edge-based network intelligence.

Accordingly, Section 2.4.2 now includes an additional paragraph explaining its selection.
The SAE-DNN was chosen for its stronger representational capacity while maintaining manageable computational costs. In contrast to convolutional or transformer-based architectures—which demand substantially more parameters and FLOPs—the DNN provides sufficient depth to capture nonlinear feature interactions within the SAE’s latent space, making it well suited for edge environments. This design achieves high classification accuracy without exceeding the memory and latency limits of devices such as the Jetson Nano and Coral Dev Board. 
As such, SAE-DNN represents the upper bound of achievable accuracy within the proposed efficiency-oriented framework, serving as a balanced counterpart to the lightweight SAE-FF model.

Comments 8: To ensure reproducibility, enhance the experimental setup description with comprehensive details, including exact software versions and all precise hyperparameters used for model training.
Response 8: We appreciate the reviewer’s suggestion. Section 2.7 (Implementation Environment) has been expanded to include additional details about the experimental configuration. Furthermore, additional training details were included, specifying the optimizer, learning rate, batch size, number of epochs, and regularization settings.
All models were trained using the Adam optimizer with a learning rate of 1×10⁻³ and a batch size of 1024. The number of training epochs was set to 30 for the SAE, 20 for the FF classifier, and 30 for the DNN classifier. A fixed random seed (42) was applied in all experiments to ensure reproducibility.
The sparse autoencoder employed a hidden dimension of 4, a sparsity objective (ρ = 0.05), and a penalty factor (β = 1×10⁻³). Both the SAE-FF and SAE-DNN models used the ReLU activation function, while the DNN additionally applied dropout (p = 0.2) for regularization. All experiments were conducted using 5-fold cross-validation under identical preprocessing and parameter configurations.

Comments 9: The analysis of your findings must be more insightful, including a cost-benefit analysis comparing SAE-FF and SAE-DNN, weighing accuracy against computational costs.
Response 9: We appreciate the reviewer’s valuable comment. The Discussion section (Section 4) has been significantly expanded to include a detailed cost–benefit analysis comparing SAE-FF and SAE-DNN in terms of accuracy, FLOPs, latency, and power consumption. This revision clarifies the trade-offs between performance and efficiency and interprets their implications for edge-AI deployment. The updated content appears on pages 18, Section 4 (Discussion).

Comments 10: You must acknowledge the limitations of your study (e.g., dataset scope, lack of real-world traffic scenarios) with greater transparency, discussing how these constraints impact generalizability. Based on these limitations and core results, suggest clear and promising directions for future research.
Response 10: We sincerely thank the reviewer for this helpful comment. In response, the Conclusions section (Section 5) has been revised to more clearly describe the study’s main limitations, such as the use of a single dataset (CIC-IDS2017), the binary classification setting, and the partial estimation of latency and power measurements. 
These factors are now discussed in relation to how they may limit the generalizability of the proposed framework. In addition, a paragraph has been added outlining future research plans, including validation on diverse datasets (UNSW-NB15, Bot-IoT, ToN-IoT), the application of federated and continual learning, and model-level optimization techniques such as quantization, TensorRT, TFLite, and Edge TPU deployment.
These revisions improve the clarity of the Conclusions section and better connect the study’s findings to future research directions toward practical edge-AI applications.  

Reviewer 3 Report

Comments and Suggestions for Authors

ِTher paper is intersted and hot topic, but there are some minor comments :

Accuracy equation  should be (TP + TN) / (TP + TN + FP + FN).

Authors compared there algorthm  to other AEs in the related work. It could be better to compare that with other common feature reduction techniques.

The Recall equation should be TP / (TP + FN)

It will be nice to add Edge Performance Efficiency Score to the keywords

Author Response

Comments 1:  Accuracy equation should be (TP + TN)/(TP + TN + FP + FN). The Recall equation should be TP/(TP + FN).
Response 1: Thank you for pointing this out. I've revised the formula in Section 2.8 (Table 9, page 13) to clearly indicate the standard definition in parentheses for clarity.

Comments 2:  Authors compared their algorithm to other AEs in the related work. It could be better to compare that with other common feature reduction techniques.
Response 2: We agree with the reviewer. To address this suggestion, a new paragraph was added at the end of Section 1.2.1 (p. 3, lines 89-101) comparing SAE with PCA, LDA, t-SNE, and UMAP. This addition clarifies why SAE was selected for non-linear traffic features and real-time edge inference.

Comments 3:  It will be nice to add Edge Performance Efficiency Score to the keywords.
Response 3: We have added “Edge Performance Efficiency Score (EPES)” to the list of keywords to highlight the evaluation metric proposed in our study.
The updated keywords are as follows: 
Keywords: edge computing; sparse autoencoder; lightweight classifier; network anomaly detection; feed-forward neural network; deep neural network; CIC-IDS2017; Edge Performance Efficiency Score (EPES)

Round 2

Reviewer 1 Report

Comments and Suggestions for Authors

This version is well-written and can be published. 

Author Response

Comments 1: "This version is well-written and can be published."
Response 1: We sincerely thank Reviewer #1 for their positive and encouraging assessment.

Comments 2: 'Results' and 'Conclusions' sections marked as 'Can be improved'.
Response 2: Response 2: We appreciate this valuable suggestion. In response, we have thoroughly revised the Results, Discussion, and Conclusions sections to enhance their analytical depth and to better connect our findings with their broader implications.

Reviewer 2 Report

Comments and Suggestions for Authors

Dear Authors,

I have reviewed the revised manuscript and find that the authors have satisfactorily addressed most of the concerns raised in the initial review. The improvements in clarity, structure, and methodological justification are evident and appreciated.
I recommend acceptance with minor revisions. The remaining issues relate to the presentation of tables and figures. Specifically, the tables should be adjusted to improve readability, and the figures require better resolution and visual clarity to meet publication standards.

Author Response

Comments 1: The Introduction requires improvement in its background and references.
Response 1: Following the reviewer’s helpful advice, we have strengthened the Introduction by incorporating two recent and highly relevant references ([20] and [21] in the revised manuscript). These additions help to more clearly position our study within current research trends and reinforce the motivation behind our approach.

Comments 2: The tables should be adjusted to improve readability, and the figures require better resolution and visual clarity 
Response 2: We fully agree. All figures have been re-generated at 300 DPI resolution for improved visual quality. In addition, all tables were reformatted to enhance structure, consistency, and readability in line with MDPI formatting standards.

Comments 3: The Research design and Conclusions sections were marked as 'Can be improved'.
Response 2: To address this comment, we have expanded the Discussion and Conclusions sections. These revisions provide a more comprehensive interpretation of the experimental findings and emphasize the practical implications of our work. We believe this strengthened analysis now fully supports the research outcomes and aligns with the reviewer’s expectations.
